# Integration of In Silico and In Vitro Analysis of Gliotoxin Production Reveals a Narrow Range of Producing Fungal Species

**DOI:** 10.3390/jof8040361

**Published:** 2022-03-31

**Authors:** Sergio Redrado, Patricia Esteban, María Pilar Domingo, Concepción Lopez, Antonio Rezusta, Ariel Ramirez-Labrada, Maykel Arias, Julián Pardo, Eva M. Galvez

**Affiliations:** 1Instituto de Carboquımica ICB-CSIC, 50018 Zaragoza, Spain; sergio-redrado@hotmail.com (S.R.); mpdomingo@icb.csic.es (M.P.D.); 2Biomedical Research Centre of Aragon (CIBA), Fundacion Instituto de Investigacion Sanitaria Aragon (IIS Aragon), 50009 Zaragoza, Spain; patri.estb@gmail.com (P.E.); aramirezlabrada@yahoo.es (A.R.-L.); maykelariascabrero@gmail.com (M.A.); pardojim@unizar.es (J.P.); 3Department of Microbiology, Hospital Universitario Miguel Servet, IIS Aragón, 50009 Zaragoza, Spain; concepta963@gmail.com (C.L.); arezusta@unizar.es (A.R.); 4Department of Microbiology, Pediatrics, Radiology and Public Health, University of Zaragoza, 50009 Zaragoza, Spain; 5Aragon I+D Foundation (ARAID), 50018 Zaragoza, Spain

**Keywords:** gliotoxin, bisdethiobis(methylthio)gliotoxin, NRPS, epipolythiodioxopiperazines, gli cluster

## Abstract

Gliotoxin is a fungal secondary metabolite with impact on health and agriculture since it might act as virulence factor and contaminate human and animal food. Homologous gliotoxin (GT) gene clusters are spread across a number of fungal species although if they produce GT or other related epipolythiodioxopiperazines (ETPs) remains obscure. Using bioinformatic tools, we have identified homologous gli gene clusters similar to the *A. fumigatus* GT gene cluster in several fungal species. In silico study led to in vitro confirmation of GT and Bisdethiobis(methylthio)gliotoxin (bmGT) production in fungal strain cultures by HPLC detection. Despite we selected most similar homologous *gli* gene cluster in 20 different species, GT and bmGT were only detected in section Fumigati species and in a *Trichoderma virens* Q strain. Our results suggest that in silico gli homology analyses in different fungal strains to predict GT production might be only informative when accompanied by analysis about mycotoxin production in cell cultures.

## 1. Introduction

Gliotoxin (GT) is a fungal toxin belonging to the family of epipolythiodioxopiperazines (ETPs) which possesses antimicrobial and immunomodulatory functions [1]. Its chemical structure is defined by the presence of a transannular disulfide bond, formed by the addition of two sulfur atoms to the cyclic bond of amino acids L-Ser and L-Phe. In fact, the high reactivity of this chemical group confers GT the ability to cross-link other biological molecules through formation of a disulfide bridge and to generate reactive oxygen species by redox cycling [2,3]. Indeed, most of its biological roles are due to its structure and reactivity.

GT has been found to exert biocidal activity against a large number of species belonging to almost all kingdoms of life, and also against some viruses. In case of bacteria, GT inhibits in vitro growth of *Mycobacterium tuberculosis*, *Pseudomonas aeruginosa*, *Enterobacter aerogenes*, *Escherichia coli*, *Microsporum gypseum*, *Staphylococcus aureus,* and *Bacillus subtilis* [4,5,6]. In regard to the kingdom of fungi, GT has been reported to prevent in vitro growth of human pathogen *C. albicans* [7]. Protozoa such as *P. falciparum*, which is responsible for malaria disease, are also susceptible to the presence of GT [8]. Furthermore, antiviral activity of GT against a wide range of viruses such as Nipah virus, Hendra virus, polio virus, herpes simplex virus, coxsackie virus or influenza A virus [9,10,11,12] among others has been described.

A separated mention is required for GT roles in animals, especially mammals. GT is secreted during infection by *A. fumigatus* in order to avoid host immunity and help colonization. Indeed, GT is known to be a virulence factor of this fungus that enhances invasion of mouse lungs, leading to an increased mortality [3,13,14,15]. GT is secreted during the first steps of conidial germination and hyphae invasion because it can be detected in lungs from infected mice since day 1 post infection [16]. Secreted GT might play a critical role in fungal survival because it inhibits macrophage phagocytosis, causes apoptotic cell death in macrophages and monocytes, blocks cytokine production through inhibition of transcription factor NF-κB, and prevent degranulation of mast cells [3,17,18,19,20,21,22,23]. Apart from its role as virulence factors and its potential function during host invasion, GT has been shown to induce apoptosis in a vast range of tumoral cells such as mouse L929 fibroblast cells, human and rat hepatic stellate cells, human cervical cancer cells or colorectal cancer cells [24,25,26,27]. Thus, it has been proposed as a potential anti-cancer drug, albeit due to its toxicity against healthy tissues/cells, and selective ways of delivery to cancer cells will be required before it could be tested in vivo like it was recently proposed [28]. In addition, due to its high biological activity and potential harmful effects against mammals and other species, GT contamination of human and animal food has raised several concerns in recent years [29,30,31].

Like many other fungal secondary metabolites, enzymes that synthetize GT are encoded in a gene cluster. *A. fumigatus* GT gene cluster (*gli* cluster), which is one of the most studied, contains 13 genes needed for GT biosynthesis [32] (Figure 1). At one end of the gene cluster, it is placed *gliZ*, which is a zinc finger transcription factor responsible for the expression of several genes into the cluster [15]. One of those genes is *gliP* that encodes a non-ribosomal peptide synthase (NRPS) responsible for the first step in GT biosynthesis. GliP is a three-module NRPS (A1-T1-C1-A2-T2-C2-T3) which is capable of synthesizing a cyclic peptide by linking L-Phe and L-Ser [33]. Although it lacks a thioesterase domain that enables the releasing of the cyclopeptide, in vitro studies using a purified enzyme have shown that this happens slowly using a wild type *A. fumigatus* Af293 GliP [33].

Once the diketopiperazine backbone is synthetized, cytochrome P450 GliC catalyzes the next step by hydroxylation of both old L-Phe and L-Ser alpha carbons [35]. Two sulfur atoms are added subsequently by the substitution of the recently bonded hydroxyl groups by two glutathione molecules. This reaction is carried out by the enzyme GliG [36]. Complete removal of glutation molecules is performed by three proteins named GliK, GliJ, and GliI [37]. GliK, a gamma glutamyl cyclotransferase, releases glutamyl moieties from glutathione linked to the backbone; GliJ, a dipeptidase, cleaves glycines forming the remaining glutathione; and GliI, a lyase, breaks the C-S bond completely eliminating the remaining moiety from the backbone [37]. In this way, the released product (3-benzyl-3,6-dithio-6-(hydroxymethyl)-diketopiperazine or dithiol gliotoxin) contains the two characteristic thiol groups of GT.

Next step in GT biosynthesis is considerably important. As thiol groups can react with many other molecules, GliT (thioreductase) oxidizes both groups by forming a disulfide bond and minimizing self-toxicity induced by this GT intermediate [38,39,40]. Dithiol GT is also a substrate for GtmA, to be converted to Bisdethiobis(methylthio)gliotoxin (bmGT) [41,42]. Once GT is produced it will induce *gli* cluster expression [43] or be secreted from *A. fumigatus* via GliA [44]. Due to the high GT reactivity *A. fumigatus* has developed self-protection mechanisms being the two most important, bmGT formation that will inactivate the toxin and secretion that will reduce *gli* expression and GT production [41]. 

Several fungal species have been suggested to synthetize GT, such as *Trichoderma lignorum* [45], *Trichoderma virens* [46], *Penicillium obscurum* [46], *Gliocladium fimbriatum* [47], *Candida albicans* [48], and different *Aspergillus* spp., such as *A. fumigatus* [49], *A. terreus*, *A. niger,* and *A. flavus* [50], among others. However, contradictory findings have been reported regarding GT synthesis in some of these fungal families [46,50,51,52,53]. Although there are differences between the gene clusters involved in GT synthesis depending on the fungal species that possess it, such as the number of genes that it contains or their synteny and disposition, there are also similarities between them as we expose in this work. Since GT production by different fungal species might be relevant for diverse fields including heath, food, and agriculture, it would be important to know which environmental fungus might produce GT.

Here we apply different bioinformatic strategies to search and study protein structures of several fungal species most similar to *gli* cluster and concretely *gliP* of *A. fumigatus*. We start from the fact that high sequence similarity provides a similar three-dimensional protein structure, which is a determining factor in its biological activity. Thereby, the alleged GT production in these species could be encoded by the same or similar gene cluster or perhaps similar genes. In order to shed some light on the controversy that currently exists about which fungal species are GT producers, an extensive study of the species sequences was carried out using different bioinformatic tools. In silico studies empower us to find similar genetical or proteinic sequences among all the species whose genomes have been sequenced, and ease the search of these sequences and allow inferring that the sequences of different species that share a high grade of similarity can retain the same function. However, the bioinformatic study must be complemented with the in vitro detection and quantification of GT production in specific selected strains in order to confirm if in silico predictions were accurate.

## 2. Materials and Methods

### 2.1. Bioinformatic Study

#### 2.1.1. Search for *A. fumigatus* Af293 GliP Homologous Proteins in Other Species

The search of *A. fumigatus* Af293 GliP homologous genes and proteins in other related species was done using online BLASTn and BLASTp [54] (https://blast.ncbi.nlm.nih.gov/Blast.cgi, accessed on: 26 March 2020) tools. 

Selection of the possible GliP homologous proteins in other species was defined by query coverage (>90%) and aminoacidic sequence identity (>35%). After the quick search, we picked 22 homologous GliP sequences belonging to different fungal species. A multiple sequence alignment was performed with the selected sequences by using Clustal Omega (http://www.clustal.org, accessed on: 26 March 2020), an online tool that uses seeded guide trees and HMM profile-profile techniques to generate alignments between three or more sequences [55].

To easily understand the results, sequence alignments were visualized with Jalview [56] software. Jalview software can show graphically the conserved residues and regions between the different NRPSs selected.

#### 2.1.2. Phylogenetic Tree

The phylogenetic analysis helps to understand the functional roles of conserved domains in sequences. In this work we performed a phylogenetic analysis of the proteins previously selected by the best BLAST results, similar to NRPS of *A. fumigatus* Af293. A phylogenetic tree was obtained by using Molecular Evolutionary Genetics Analysis across computing platforms (MEGA X) software [57]. This analysis involved 22 amino acid sequences. There were a total of 2379 positions in the final dataset. The Maximum Likelihood method and Le_Gascuel_2008 [58] model was selected to infer the evolutionary history. The goodness-of-fit of various models to our data was measured, and finally Le_Gascuel_2008 model was chosen based on Bayesian information criterion (BIC) [59] and Akaike information criterion (AIC) [60] values. A discrete Gamma distribution was used to model evolutionary rate differences among sites (+G, parameter = 1.0608). The rate variation model allowed some sites to be evolutionarily invariable ([+I], 5.76% sites).

MEGA X software finds an initial tree obtained automatically by applying Neighbor-Join and BioNJ algorithms to a matrix of pairwise distances estimated using the Jones-Taylor-Thornton (JTT) model. Then, heuristic search is performed and the topology with superior log likelihood value is selected. 

One of the most used tests of the reliability of an inferred tree is Felsenstein’s [61] bootstrap test, which is evaluated using Efron’s [62] bootstrap resampling technique. This bootstrap method was performed with 100 replications. The bootstrap values in each clade indicate how many times out of 100 (in our case) the same grouped taxa are found when repeating the phylogenetic reconstruction on a resampled dataset. Generally, if the bootstrap value for a given interior branch is 95% or higher, then the topology at that branch is considered correct [63].

#### 2.1.3. Search for Possible Clusters of Secondary Metabolites in Various Strains Using antiSMASH

##### Identification of the Rest of *gli* Genes Surrounding the Selected NRPSs

antiSMASH 6.0 [64] is an online and offline software that allows the identification of all NRPS and PKS gene clusters contained in a complete bacterial, fungal or plant genome. antiSMASH uses a rule-based approach to identify those possible secondary metabolite gene clusters in a genetic sequence [64]. It first verifies the presence of NRPSs, PKSs, or other core enzymes. After NRPS/PKS identification step, the software is capable of predicting the different domains that contain those core enzymes. As an additional feature, antiSMASH offers the possibility of detecting other side genes involved in the secondary metabolite biosynthesis as well as gene cluster borders.

In this study, antiSMASH has been used to check that the previously identified NRPSs contain the same domains as *A. fumigatus* Af293 GliP. antiSMASH was also useful to recognize and annotate the nearby genes that are next to the NRPSs and whose roles can be linked to them and finally, in obtaining complete sequences of the proposed gene clusters.

##### Synteny and Gene Location Inside the Clusters

Synteny traditionally is defined as the presence of two or more similar genes in the same chromosome [65]. In this work, we only study the presence of those homologous genes contained in the gene clusters that we previously identified. Presence, position, and orientation of the several genes that are part of the antiSMASH detected gene clusters were easily and visually checked through Gene Graphics webtool [66].

##### Homology between *A. fumigatus* Af293 *gli* Genes and Their Counterparts in the Other Fungi Studied

After the identification of the possible genes included in the fungal gene clusters studied, we proceed to assess the degree of similarity between the *A. fumigatus* Af293 *gli* genes and their homologs in the other fungi. Gli aminoacidic sequences obtained from NCBI GeneBank [54] were compared through PSI-BLAST in a restricted search to the studied organisms to reveal if their best BLAST result for a determined fungus is the homologue gene contained in the predicted gene cluster.

### 2.2. Analysis and Quantification of GT Production by Several Fungal Strains

#### 2.2.1. Fungal Strains and Culturing Conditions

*A. fumigatus* 1631562 and *A. lentulus* 353 were provided by the microbiology service of the Miguel Servet hospital. *A. fumigatus* B5223 and *P. expansum* MD-8 were kindly provided by J. A. Sugui and A. R Ballester respectively. *A. pseudofischeri* CBS 404.67, *A. fischeri* CBS 420.96, *A. turcosus* CBS 140371, *Rhizodiscina lignyota* CBS 133067, *Trichoderma parareesei* CBS 125925, *Colletotrichum fructicola* CBS 120005, *Penicillium flavigenum* CBS 110407, *Trichoderma harzianum* CBS 226.95, *Trichoderma reesei* CBS 383.78, *Penicilliopsis zonata* CBS 506.65, *Trichoderma virens* CBS 249.59 and *Elsinoe ampelina* CBS 208.25 were purchased at Fungal Biodiversity Centre (CBS) in the Netherlands.

Fungal strains were cultured in Sabouraud glucose agar with 50 mg/L chloramphenicol (Merk, Darmstadt, Germany) for 7 days at different conditions for each species in order to produce a significant amount of conidia as described in Table 1. Spores were harvested with a swab smeared in Tween-20 (Thermo-Fischer, Waltham, MA, USA) and suspended in sterile water. Conidial suspensions were adjusted to 2 McF and 1 mL was poured into cell culture flasks containing 9 mL of Czapek-Dox broth (Thermo-Fischer, Waltham, MA, USA) or 9 mL of Roswell Park Memorial Institute 1640 broth (RPMI 1640) with 20 mM 4-(2-hydroxyethyl)-1-piperazineethanesulfonic acid (HEPES) and without NaHCO_3_ (Merk, Darmstadt, Germany) and cultured for 4 days at 37 °C. After 4 days, supernatants were filtered with a 0.22 µm filters and stored at −20 °C until further uses. All fungal strains were cultured in Bsl-2 lab facilities at IIS Aragón. If a species, such as *A. lentulus* or *E. ampelina*, could not sporulate in the indicated time or media, a small piece of 1 × 1 cm of mycelia was sliced from the agar plate and inserted into cell culture flasks. 

#### 2.2.2. GT and bmGT Chemical Extraction

A total of 3 mL of fungal culture supernatant was mixed with 10 mL of dichloromethane (DCM) into a 20 mL glass jar and vortexed for 30 s. Organic phase containing GT was transferred to another jar and evaporated under nitrogen flux. The solid residue was dissolved in 1 mL of DCM and evaporated in a 2 mL glass vial under nitrogen flow again. Solid residue was dissolved in 55% MiliQ water and 45% methanol prior to HPLC analysis.

#### 2.2.3. HPLC Analysis and Quantification

High-performance liquid chromatography (HPLC Alliance e2695, Waters, Milford, MA, USA) was used for the quantification of GT in culture extracts as described before [50,67,68,69]. A C-18 column (XBridge^®^ C18 3.5 µm 4.6 × 100 mm Column, Waters) was used for setting the temperature at 30 °C. HPLC analysis of samples was performed as gradient elution using water and methanol: 0–10 min: 55% water; 10–11 min: 40% water; 11–20 min: 40% water; 20–22 min: 55% water. The flow rate was set to 0.8 mL/min and the injection volume was 10 µL. An ultraviolet signal detector (2489 UV/Vis Detector, Waters) was used, monitoring the absorbance at 273 nm. A standard curve was obtained with a GT standard ranging from 0.1 to 35 µg/mL (Appendix A). As a positive control, a GT and bmGT standard extracted from RPMI or Czapek-Dox medium was used. The overall recovery was dependent on the starting concentration, reaching up to 72% and 91% recovery of GT and bmGT, respectively. The detection limit of the HPLC was 16.7 ng/mL for both GT and bmGT.

## 3. Results

### 3.1. Bioinformatic Study

#### 3.1.1. Search for *A. fumigatus* Af293 GliP Homologous Proteins in Other Species 

About a third of the species obtained by BLAST belong to the section *Fumigati*, there are two *penicilliums*, four *trichodermas*, as well as other less common species such as *coleophoma* or *colletotrichum*.

NRPS with high sequence homology can be expected to be orthologous to GliP performing similar biosynthetic functions, or even the same as GliP. Assuming homology, it is also likely that the sequences and functions of the rest of the auxiliary enzymes encoded in the genes of the cluster are conserved. In this study, we have manually selected 20 NRPS from different species with a relatively high degree of similarity (query coverage > 90%, aminoacidic sequence identity > 35% and e-value = 0). 

As it is shown in Table 2, best BLAST results for *A. fumigatus* Af293 *gliP* gene searching against all fungi are always gliPs from other *A. fumigatus* strains, which validates our approach. Thus, we selected only one strain (*A. fumigatus* A1163) to assess variation within strains. Next highest scored BLAST results are given by other section Fumigati species such as *A. fischerii*, *A. turcosus*, *A. novofumigatus,* or *A. lentulus* among others. These NRPSs belonging to section Fumigati species are expected to be *A. fumigatus* GliP orthologues due to the high genetic proximity among them. It should be noted that no *Aspergillus* species outside section Fumigati are obtained as BLAST results, inferring that only Fumigati section species might be capable of producing GT. In fact, best BLAST outcomes for *A. terreus* (section Terrei) and *A. flavus* [70] (section Flavi) are respectively AtaP and AclP, which identity is fewer than 35% (data not shown). AtaP is known to be the NRPS of the acetylaranotin gene cluster and catalyzes the condensation of two L-Phe to form a cyclo-Phe-Phe [70]. The same frame is generated by AclP [71], a cyclo-Phe-Phe, but in this case, the rest of *A. flavus* GT-like cluster genes generate a different secondary metabolite known as aspirochlorine.

However, out of *Aspergillus* section Fumigati, the sequence identity drops to 60–35%, although still conserving high percentages of covered sequence. Given that ataP and aclP paralogues have a few worse scores than lower range BLAST results and differs from GliP in one adenylation domain specificity (GliP form a cyclo-Phe-Ser), it is difficult to determine if the NRPSs not belonging to the Fumigati section keep their specificity for L-Phe and L-Ser as GliP. Supporting these results, it was previously shown that *A. niger* and nidulans isolates do not produce GT, and that generation of GT in other non-fumigati sections such as terreus or flavus was limited to very few isolates. Outside the section Fumigati, the three Penicillium species achieve the best identity outcomes of around 60%. Among low rated best BLAST results, it is worth to mention *T. virens* Gv29-8 GliP, which is known to produce GT, albeit it owns just a 44% of identity [72].

The multiple sequence alignment of the 23 NRPS performed by Clustal Omega and visualized by Jalview allows to determine the sequence fragments or amino acids conserved between all the sequences. Jalview visual analysis of the alignments shows that most conserved regions across the sequences are in the adenylation (Figure 2) domains while condensation domains exhibit less homology than the others.

#### 3.1.2. Phylogenetic Tree 

The phylogenetic tree was inferred from the sequences similar to *A. fumigatus* Af293 GliP. The tree with the highest log likelihood (−52146.32) was selected (Figure 3). In most cases, strains belonging to the same fungal species have been grouped in the same clade. The clearest example is that of the genus *Aspergillus*, in the upper part of the phylogenetic tree.

The bootstrap value of the clade that encompasses all the *Aspergillus* taxa has a bootstrap value of 100. This means that the 100 times that the tree inference algorithm has been performed, these sequences have been classified in the same clade. Therefore, the topology of this branch is highly reproducible giving a high confidence value. Similarly, the proteins that have been selected to study in the Penicillium strains have been classified in the same clade with a bootstrap value of 100. In turn, these proteins come from the same common ancestor as those of the *Aspergillus genus*, with a bootstrap value of 100. This result suggests high similarity between the sequences of these proteins. 

The branch length among *A. fumigatus* strains is 0, which means that there is no substitution per site from one protein to another. Between the strains of *A. fumigatus* and the rest of the species of the genus *Aspergillus,* we found small distances (0.02–0.06). On the other hand, among the *Penicillium* strains, the distance is slightly higher (0.1–0.2), showing more evolutionary differences between them. Besides, the length of the branch between *Aspergillus* and *Penicillium* is 0.3–0.35 substitutions per site, showing the slight differences between these proteins.

Generally, the strains of the genus *Trichoderma* have also been grouped together with the same common ancestor, but in this case, the evolutionary distances are greater; therefore, we can deduce that these proteins present more differences between them and with respect to the *Aspergillus genus*. Generally, the rest of the strains belonging to the same species have also been grouped together by the algorithm, such as *Colletotrichum*.

The bootstrap values for most of the generated clades are 100 or close to 100, demonstrating high reliability of the created tree.

#### 3.1.3. Search for Possible Clusters of Secondary Metabolites in Various Strains Using antiSMASH

NRPSs are multimodular enzymes. *A. fumigatus* Af293 GliP is composed of two adenylation domains, two condensation domains, and three thiolation domains showing the structure A1-T1-C1-A2-T2-C2-T3 [33]. antiSMASH analysis of the fungal NRPSs studied here reveals that most of them conserve all these domains. However, as it is represented in Figure 4, several NRPSs sequences lack one or more domains. This is the case of *B. victoriae* FI3 EUN32662.1, *C. fructicola* CGMCC3.17371, *C. asianum* ICMP 18580 KAE9572546.1 proteins, that according to antiSMASH, lack T3 domain which can lead to a diminished cyclization ability of the NRPS [73]. Other NRPSs such as *C. cylindrospora* BP6252 RDW70942 do not contain T1 and T3 domains, which could suggest absence of the function performed by these enzymes. However, PFAM analysis through antiSMASH could not recognize T1 and T3 domains in *T. virens* Gv29-8 EHK22005, which is known to be a functional enzyme and synthetize cyclo-Phe-Ser [72], which suggest alternative protein domains that could present a similar function.

NRPS are usually included inside of a more complex gene cluster. Accompanying these NRPSs are other genes that codify enzymes that participate in the same secondary metabolite biosynthesis. In order to elucidate if selected gene clusters are capable of synthetize GT, it is necessary to identify homologs to the rest of the genes known to be involved in the synthesis. 

As it is shown in Figure 5, most *gli* clusters homologues studied here preserve the majority of those genes mentioned above. All *Aspergillus* section Fumigati species contain inside their *gli* clusters all the genes present in *A. fumigatus* Af 293 *gli* cluster, so they should be able to synthetize GT. However, *A. turcosus* HMR Af 1038 *gli* cluster lacks *gliZ* gene, which encodes the transcription factor that is responsible for the expression of several *gli* genes, suggesting that this specie would not be able to produce GT [15]. As it is shown in Appendix A, there is high identity and query coverage among Fumigati section *gli* homologs.

In case of *P. flavigenum* IBT 14082, most of the homologs *gli* genes are present within the cluster. Nevertheless, the cluster lacks *gliF* and *gliZ* and contains two N-methyltransferases (gliN). Moreover, Best Blast Hit for *gliA* homologue is a gene outside the cluster although an MFS is located within the cluster and there is other nearby. For *P. expansum* MD-8, similar results were obtained except the fact that *gliZ* is present and there are no two *gliN* homologs. As expected, lower sequence identities were obtained when *Penicillium gli* homologue genes are compared to *A. fumigatus* Af 293 *gli* genes.

*B. victoriae gli* gene cluster contains all genes involved in GT biosynthesis except *gliF* when compared to *A. fumigatus* Af 293 *gli* cluster. The only difference is that there are two *gliN* genes together. It must be mentioned that other MFS outside the cluster express more homology with *A. fumigatus* Af 293 *gliA*, rather than the contained in. Something similar happens with *A* gene in *E. ampelina* homologue gene cluster. This gene cluster contains an MFS which it is not the Best Blast Hit for *A. fumigatus gliA*, although it retains a certain grade of homology. However, the selected gene cluster contains all genes that *A. fumigatus* needs to synthetize GT, and some others extra genes with an unknown role in GT synthesis which are shown in grey color in Figure 5. It must be mentioned that a few more genes are located next to the *gliK* gene at one end of the cluster and which have related secondary metabolism functions such as cytochromes P450 or N-methylases. As in case of *E. ampelina gli*-like gene cluster, *C. cylindrospora* BP6252 *gli* homologous gene cluster has all genes needed to GT biosynthesis although *A* and *Z* genes are not Best Blast Hits in *C. cylindrospora* genome.

*Colletotrichum* species studied here possess shortened *gli* homologous gene clusters but one differs a lot from the other. *C. fructicola* CGMCC3.17371 *gli* homologous gene cluster lacks *K* and *I* genes, although a homologue of the last one resides next to a side of the gene cluster. As in other species here studied, *A* and *Z* genes are not the Best Blast Hits compared to those in *A. fumigatus*. *C. assianum* lacks most genes related to GT biosynthesis, conserving only five of those genes. It must be mentioned that from here, antiSMASH cluster prediction identifies the selected clusters as gliovirin related clusters.

*T virens* Gv29-8, which is a known GT producer (Q strain), apparently only owns 8 of 13 genes involved in GT biosynthesis contained within the GT gene cluster. However, these *gli* genes are contained in a small scaffold of about 30 Kb, so the rest of the genes might be spread throughout the rest of the genome [74]. In fact, possible candidates of *T. virens gliJ*, *gliA,* and *gliT* are located in the small scaffolds 51, 34, and 10 which have 40, 13, and 7 Kb respectively. GliZ and gliH (hypothetical protein) presence within the genome is still unknown.

As in other species here studied, *T. hemipterigena* BCC 1449 *gli* homologous gene cluster contains almost all *A. fumigatus gli* genes, with the exception of *F* gene. *The T. hemipterigena* BCC 1449 gene cluster also contains two MFS genes which exhibit low similarity with *A. fumigatus gliA*. *Z* and *T* gene are also present inside *T. hemipterigena* gene cluster but they are not the Best Blast Hit when compared with their *A. fumigatus* counterparts. *T. hemipterigena* gene cluster is identified as a gliovirin gene cluster homologue by antiSMASH again.

The last cluster represented in the Figure 5 and belonging to *P. zonata*, *R. lignyota*, and the rest of the *Trichodermas* have better resemblance to the verticillin gene cluster than GT one. Except *R. lignyota*, all the other species contains in their clusters an A protein which codifies for an ABC transporter, not an MFS as GT-like gene clusters. Other differences lie on the presence of a kind of fusion gene composed with the *Z* and *B* gene (*T. reesei*, *T. parareesei*, *T. harzianum* and *R. lignyota*) and the presence of the *L* gene similar to verL (*T. reesei*, *T. parareesei*, *T. harzianum*) or *D* gene similar to aflD (aflatoxin D biosynthetic protein).

In fact, *P* and *C* genes are, as a rule, the most conserved enzymes in all the species we have studied, and the intra-cluster synteny of these two genes is ultra-conserved. In all clusters, the genes are contiguous and in similar orientation. Other genes such as *G* and *K* generally occur together or very close to each other. In general, most of the functions required for GT biosynthesis are present in the *gli* cluster homologs studied here.

### 3.2. Analysis and Quantification of GT Production by Several Fungal Strains

To check the validity of the in silico approach, GT and bmGT (that derives from GT dimethylation) presence in fungal cell culture supernatants from selected fungal species used in the in silico study was analyzed by HPLC. As it is shown in Table 3, GT and bmGT production was only found in section Fumigati species grown in Sabouraud and RPMI broth, except for *A. lentulus* and *A. fischeri*. Outside of the Fumigati section, only *T. virens* Q strain was capable to synthetize GT and bmGT. Under these conditions the rest of the species analyzed were not able to produce GT or bmGT. GT and bmGT concentrations obtained from culture supernatants ranged from 0.085 to 11.067 µg/mL and from 0.222 to 11.131 µg/mL respectively.

## 4. Discussion

GT is a toxin and virulence factor of *A. fumigatus* that has been reported to be produced by other *Aspergillus* spp. and other fungal genus with relevance for human and animal health due to their ability to produce infections or to contaminate human and animal food. Here, based on the presence of described protein clusters involved in GT biosynthesis, we have developed a bioinformatics analysis of fungal genus and species that potentially can produce GT, which has been further validated at the lab by analyzing GT production by selected fungal species.

Phylogenetic analysis is essential for the comparative study of protein sequences. Their results have many applications in the study of the evolution and functions of proteins, as well as in the prediction of the function of the genes that encode them, the identification, construction, and discovery of gene families and the annotation of the genome [75]. For these reasons, it is more appropriate to use protein sequences to analyze the phylogeny of species than to use DNA sequences [76,77,78]. Proteins with high sequence identity tend to have evolutionary relationships and similarities in function, indicating conserved biological function [79]. Phylogenetic analysis provides results represented by a phylogenetic tree, in which sequences are grouped based on sequence similarities.

The fact that the proteins of the strains of the *Aspergillus* and *Penicillium* genera are located evolutionarily close in the phylogenetic tree corroborates the similarity of sequences that we already obtained with the BLAST analysis. For the rest of the proteins there is also coherence between the percentage of query coverage, identity obtained from BLAST and the taxonomy of the inferred phylogenetic tree.

The conserved domains (A1-T1-C1-A2-T2-C2-T3) in *A. fumigatus* GliP registered in NCBI [80] are also found in most of the other NRPS of the studied strains. It was previously described that A1-T1-C1-A2-T2 are essential for the L-Phe-L-Ser diketopiperazine (DKP) formation and C2 and T3 domains were apparently not necessary for the DKP formation given that under the reaction conditions that Balibar and Walsh performed, the release of the DKP happened spontaneously [33]. However, it was recently described that in vivo C2 and T3 deletions disrupt in vivo GT biosynthesis since only small amounts of GT were discovered in truncated *A. fumigatus* cultures [73]. In agreement with these results, here we were unable to detect GT or bmGT in *C. cylindrospora* and *C. fructicola* cultures. Their NRPSs lacks certain domains, such as T3 and T1, which could suggest the in vivo absence of production of cyclo-L-Phe-L-Ser and thus the disruption of GT biosynthesis. However, the module domains analysis of antiSMASH was also unable to detect T1 and T3 domains in *T. virens* Gv29-8 GliP, although this enzyme is known to be functional. We have found that *T. virens* CBS 249.59 is able to produce GT in cell cultures, suggesting the presence of alternative domains with a similar function. 

Assuming that homologous *P* genes in all species studied here were functional, another reason for not having detected GT and bmGT in their cultures could be differences in the specificity of their adenylation domains. First adenylation domain of GliP can accommodate either L-Phe or L-Trp [35], so substrate specificity might have changed enough for the enzymes studied in this work as to produce different ETPs. Further research on this field is needed to confirm these hypotheses.

As expected, almost all species of section Fumigati were able to produce GT and bmGT since they preserve the entire cluster. As an exception, *A. turcosus* HMR Af 1038 genome lacks gliZ transcription factor, which would indicate that it is not capable of producing GT [15] or that its production is greatly diminished [81]. However, culture supernatants of *A. turcosus* CBS 140371 contained significant amounts of GT. Pending experimental validation, we speculate that GT production could be due to possible differences between the *gli* cluster of the strains, the presence of an unidentified copy of *gliZ* outside the *gli* cluster or the absorption of the function by other transcription factor. 

On the other hand, *A. lentulus* apparently owns all the machinery needed for GT biosynthesis when compared with *A. fumigatus*, although we could not detect any trace of GT or bmGT under our experimental conditions. Previous studies have demonstrated that a very low frequency of *A. lentulus* strains and isolates are capable to generate these secondary metabolites [53]. It must be mentioned that other studies were not able to detect these toxins in other *A. lentulus* strains [52,82]. Although it is clear that *A. lentulus* genome contains the entire *gli* cluster, further research is needed to elucidate why the secondary metabolite profiles of *A. lentulus* are very different from those of *A. fumigatus* as previously shown [52,82] despite being species closely related to *A. fumigatus*. Probably, some species that apparently owns all the machinery needed for GT biosynthesis failed to produce GT in the research due to culture conditions, since it is well-known that cryptic genes are repressed in normal culture conditions.

Although the non-pathogenic fungus *A. fischeri* had been described as not capable of producing GT [51,83], recent work shows that it does [84]. While *Knowles* et al. discovered that culturing conditions that induce GT synthesis in *A. fumigatus* also induce similar amounts of GT in *A. fischeri* cultures, our results for *A. fischeri* CBS 420.96 cultures suggest that very low amounts of GT are obtained when compared with *A. fumigatus*.

Outside the section Fumigati, we have only found GT and bmGT in culture filtrates of *T. virens* CBS 249.59. However, it has been referred for a long time that some *Penicillium* species can produce GT [85,86,87,88,89]. It is possible that our culture conditions would not be the most optimal ones, but in this study, we have not obtained GT or bmGT in the two *Penicillium* species which own the most similar *gli* cluster compared to the *gli* cluster of *A. fumigatus*. It is true that *P. lilacinoechinulatum* possesses a similar *gli* cluster uploaded on the NCBI, but the absence of a complete genome on this web led us not to include it in our study. In the same way, the absence of a complete genome uploaded on NCBI of all *Penicillium* species referenced above that can produce some type of GT impede a complete understanding about the ability to produce GT in the genus *Penicillium*.

The *gli* cluster of *T. virens* Gv29-8 was previously reported to contain only 8 of 13 genes [72], here we report the possible candidates for *gliJ*, *gliA,* and *gliT* in *T. virens* genome. Since a part of the *gli* cluster of *T. virens* Gv29-8 resides at the end of a scaffold, the rest of the genes should be located at one or more scaffolds. We found that these three missed genes could be located inside other little scaffolds which also contain just a few more genes. Nevertheless, a genome sequencing of more *T. virens* strains would clarify the complete structure of the *gli* cluster of *T. virens*.

## 5. Conclusions

In silico studies provide the possibility of detection of a number of homologs gli clusters in the fungi kingdom. However, it is difficult to determine whether those homologs are capable of biosynthesizing GT, other ETPs or are otherwise functionless. Under the conditions carried out along this work, we have only achieved GT and bmGT production in culture supernatants of section Fumigati species and *T. virens*.

## Figures and Tables

**Figure 1 jof-08-00361-f001:**
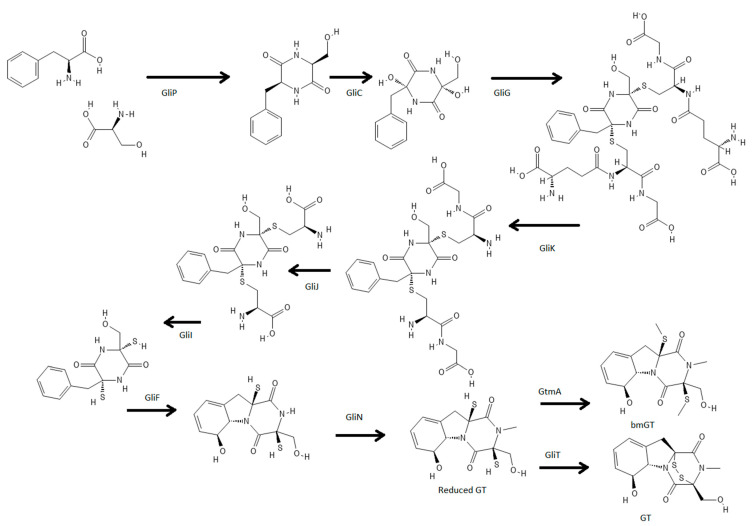
GT biosynthesis and role of the genes involved in this process [34].

**Figure 2 jof-08-00361-f002:**
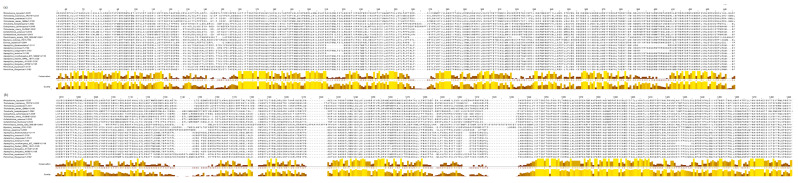
Protein sequence alignment of (**a**) first adenylation domains belonging to the fungal NRPSs studied and (**b**) second adenylation domains for the same NRPSs. Brown to yellow bars under each residue represent conservation grade and quality for each position. Yellow bars express high degree of conservation while brown bars indicate low conservation.

**Figure 3 jof-08-00361-f003:**
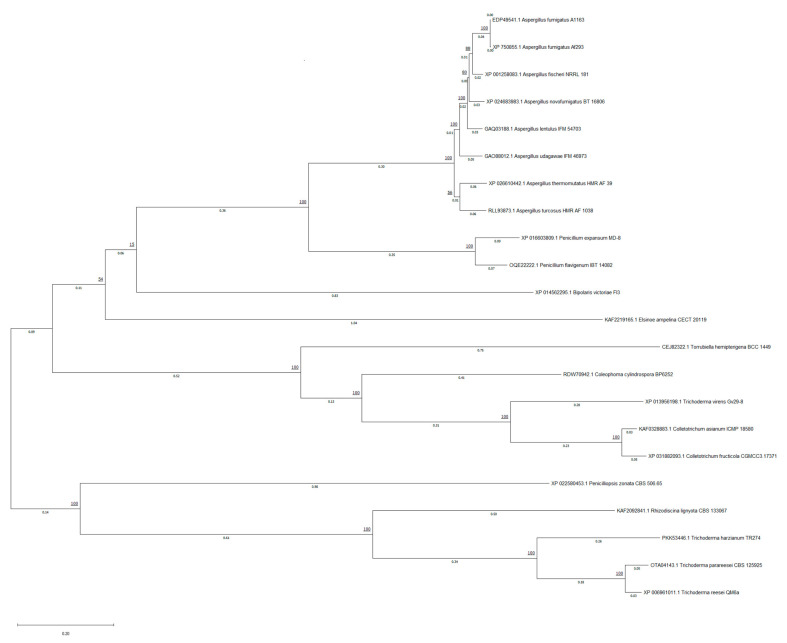
Phylogenetic tree inferred by MEGA X software. The evolutionary history is composed by 22 amino acid sequences. The tree is drawn to scale, with branch lengths measured in the number of substitutions per site (next to the branches). The percentage of trees in which the associated taxa clustered together is underlined next to each clade (bootstrap value).

**Figure 4 jof-08-00361-f004:**
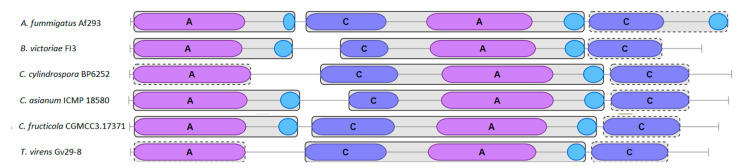
Different domains that compose the NRPS GliP and some of its homologues. Letter A represents the adenylation domains, letter C depicts the condensation domain, and blue circles show thiolation domains.

**Figure 5 jof-08-00361-f005:**
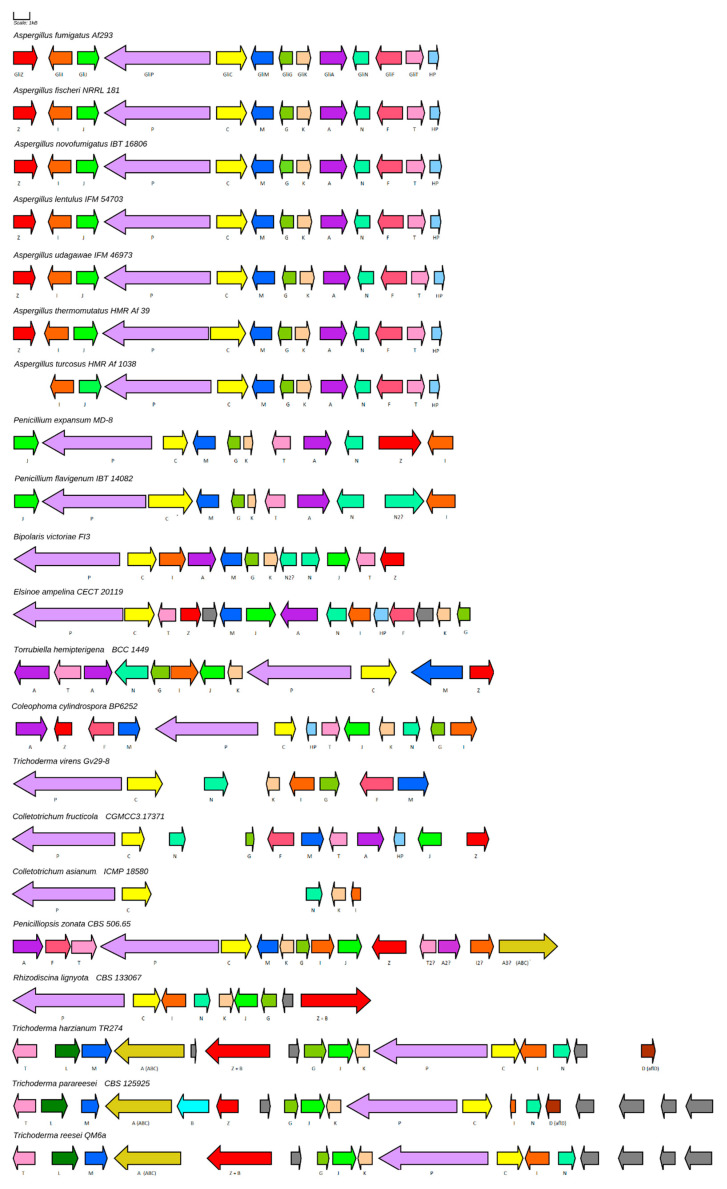
*Gli* gene cluster of section Fumigati and their homologs among the species studied. Genes that share the same color are homologues. Genes colored in grey are unrelated genes or with unknown function. Genes named as *Z* + *B* are probably product of a fusion of genes with *Z* and *B* functions.

**Table 1 jof-08-00361-t001:** Temperatures at which the studied strains were grown. If nothing is specified, strains were grown in the dark. However, *Trichoderma* genus strains were cultured under daylight at room temperature. Liquid conditions are the same for both RPMI 1640 and Czapek-Dox broth. RT = room temperature.

Fungal Strain	Solid Culturing Conditions	Liquid Culturing Conditions
*A. fumigatus* B5233	37 °C	37 °C
*A. fumigatus* 1631562	37 °C	37 °C
*A. fischeri* CBS 420.96	37 °C	37 °C
*A. lentulus* 353	37 °C	37 °C
*A. turcosus* CBS 140371	37 °C	37 °C
*A. pseudofischeri* CBS 404.67	37 °C	37 °C
*P. flavigenum* CBS 110407	30 °C	30 °C
*P. expansum* MD-8	30 °C	30 °C
*B. victoriae* CBS 174.57	30 °C	30 °C
*E. ampelina* CBS 208.25	30 °C	30 °C
*C. cylindrospora* CBS 449.70	30 °C	30 °C
*C. fructicola* CBS 120005	30 °C	30 °C
*T. virens* CBS 249.59	RT, daylight	RT, daylight
*P. zonata* CBS 506.65	RT, daylight	RT, daylight
*T. reesei* CBS 383.78	RT, daylight	RT, daylight
*R. ligniota* CBS 133067	30 °C	30 °C
*T. harzianum* CBS 226.95	RT, daylight	RT, daylight
*T. parareesei* CBS 125925	RT, daylight	RT, daylight

**Table 2 jof-08-00361-t002:** 21 best BLAST results for *A. fumigatus* Af293 GliP protein. Entries are sorted by score although query coverage, identity and e-value are also shown. These results are *A. fumigatus* Af293 itself and other *A. fumigatus sensu stricto*, several *Aspergillus fumigati* section such as *A. fischerii, A. novofumigatus or A. turcosus*, 4 *Trichoderma* species, 2 *Penicillium* species, and other less related species from genera such as *Coleophoma, Bipolaris,* or *Rhizodiscinia* among others.

Fungal Strain	Locus Tag	Accession	Score	Query Coverage	E Value	Identity
*Aspergillus fumigatus* Af293	AFUA_6G09660	EAL88817	4396	100%	0.0	100.00%
*Aspergillus fumigatus* A1163	AFUB_075710	EDP49541	4386	100%	0.0	99.77%
*Aspergillus fischerii* NRRL 181	NFIA_055350	XP_001258083	4152	100%	0.0	94.75%
*Aspergillus lentulus* IFM 54703	ALT_0509	GAQ03188	4101	100%	0.0	93.13%
*Aspergillus novofumigatus* IBT 16806	P174DRAFT_511160	XP_024683983	4012	100%	0.0	92.06%
*Aspergillus udagawae* IFM 46973	AUD_6972	GAO88012	3920	100%	0.0	89.30%
*Aspergillus turcosus* HMR AF 1038	CFD26_102651	RLL93873	3896	100%	0.0	88.42%
*Aspergillus thermomutatus* HMR Af 39	CDV56_101444	XP_026610442	3779	100%	0.0	87.07%
*Penicillium flavigenum* IBT 14082	PENFLA_c013G03821	OQE22222	2652	99%	0.0	60.95%
*Penicillium expansum* MD-8	PEX2_011780	XP_016603809	2644	99%	0.0	61.24%
*Bipolaris vicotriae* FI3	COCVIDRAFT_32734	EUN32662	1969	95%	0.0	47.29%
*Elsinoe ampelina* CECT 20119	BDZ85DRAFT_305103	KAF2219165	1712	99%	0.0	44.93%
*Coleophoma cylindrospora* BP6252	BP6252_07505	RDW70942	1642	95%	0.0	44.21%
*Colletotrichum fructicola* CGMCC3.17371	CGMCC3_g11493	KAE9572546.1	1640	94%	0.0	44.00%
*Colletotrichum asianum* ICMP 18580	GQ607_003908	KAF0328883	1627	94%	0.0	44.09%
*Trichoderma virens* Gv29-8	TRIVIDRAFT_78708	EHK22005	1621	94%	0.0	43.99%
*Torrubiella hemipterigena* BCC 1449	VHEMI02393	CEJ82322	1718	94%	0.0	41.43%
*Penicilliopsis zonata* CBS 506.65	ASPZODRAFT_160119	OJJ45943.1	1481	96%	0.0	42.54%
*Trichoderma reesei* QM6a	TRIREDRAFT_24586	EGR52474.1	1461	98%	0.0	39.94%
*Rhizodiscina lignyota* CBS 133067	NA57DRAFT_81988	KAF2092841	1461	98%	0.0	39.34%
*Trichoderma harzianum* TR274	CI102_1861	PKK53446	1453	99%	0.0	38.55%
*Trichoderma parareesei* CBS 125925	A9Z42_0047170	OTA04143	1433	99%	0.0	39.13%

**Table 3 jof-08-00361-t003:** Different concentrations of GT and bmGT obtained in culture supernatants of the fungi studied. Fungal strains were cultured in both Czapek-Dox and RPMI 1640 broth and their toxin concentration was measured by HPLC. Concentrations are expressed in µg/mL or ND if not detected. n = 3.

Fungal Strain	Czapek-Dox		RPMI 1640	
	GT (µg/mL)	bmGT (µg/mL)	GT (µg/mL)	bmGT (µg/mL)
*A. fumigatus* B5233	11.067 ± 14.286	2.595 ± 3.225	8.373 ± 2.663	11.131 ± 15.538
*A. fumigatus* 1631562	3.913 ± 4.278	0.845 ± 0.805	4.154 ± 7.442	0.519 ± 0.052
*A. fischeri* CBS 420.96	0.135 ± 0.077	0.222 ± 0.206	0.062	0.311 ± 0.108
*A. lentulus* 353	ND	ND	ND	ND
*A. turcosus* CBS 140371	0.085 ± 0.0539	1.215 ± 0.884	0.491 ± 0.585	1.795 ± 1.732
*A. pseudofischeri* CBS 404.67	0.840 ± 0.028	0.518 ± 0.297	0.295 ± 0.092	0.253 ± 0.226
*P. flavigenum* CBS 110407	ND	ND	ND	ND
*P. expansum* MD-8	ND	ND	ND	ND
*E. ampelina* CBS 208.25	ND	ND	ND	ND
*C. cylindrospora* CBS 449.70	ND	ND	ND	ND
*C. fructicola* CBS 120005	ND	ND	ND	ND
*T. virens* CBS 249.59	0.840 ± 0.824	0.518 ± 1.762	0.330 ± 0.178	2.417 ± 1.697
*P. zonata* CBS 506.65	ND	ND	ND	ND
*T. reesei* CBS 383.78	ND	ND	ND	ND
*R. ligniota* CBS 133067	ND	ND	ND	ND
*T. harzianum* CBS 226.95	ND	ND	ND	ND
*T. parareesei* CBS 125925	ND	ND	ND	ND

## Data Availability

Not applicable.

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
