# Peer review of "Integration of In Silico and In Vitro Analysis of Gliotoxin Production Reveals a Narrow Range of Producing Fungal Species"

_jof, 2022, doi:10.3390/jof8040361_

Round 1

Reviewer 1 Report

In the study the authors used in silico experiments and conventional in vitro analysis to determine fungal species that would produce gliotoxin. Although the in silico design and analysis are primitive, probably limited by the software tools and/or programming capacity, the attempt is definitely worthwhile and should be continued in the future studies. In the introduction the authors need to highlight the current pros, cons, challenges and trends of in silico analysis used for the identification of fungal species. In terms of discussion, it is not clear whether the authors used the in silico approach to guide the in vitro studies or the other way around.  Suggest including a flow diagram or a paragraph, clarifying the connection between the in vitro and the in silico data. This way readers could better understand how in silico could be used as a complementary tool for in vitro studies or even as a replacement of in vitro studies if that's authors' perspective.  

Author Response

We thank to the referee for his comments and constructive criticisms.

As suggested we have now incorporated in the manuscript “In silico studies empower us to find similar genetical or proteinic sequences among all the species whose genomes have been sequenced, ease the search of these sequences and allow inferring that the sequences of different species that share a high grade of similarity can retain the same function. However, the bioinformatic study must be complemented with the in vitro detection and quantification of GT production in specific selected strains in order to confirm if in silico predictions were accurate”.

Reviewer 2 Report

As i read all the manuscript, this work is good and valuable i believe this work is acceptable for publication in your journal

Author Response

Thank you very much for the encouraging comment.

Reviewer 3 Report

The authors report developing a bioinformatics analysis of fungal genus and species that potentially can produce gliotoxin (a toxin and virulence factor).

Abstract

detec-tion – write detection

Introduction

Line 102, appreciate a better word for proposed

Several fungal species have been proposed to synthetize GT, such as Trichoderma lig-

Methods

Brief about 22 amino acid sequences involved in analysis

Line 201

Do all fungal strain sporulate in Sabouraud glucose agar?

Line 206

Czapek-Dox broth was used for all fungal stains?

Line 207

Expand RPMI (and HEPES)

Line 273

Italisise A. niger. Some fungal strains throughout the article have not been italicized

Author Response

We thank for the comments and carefully editing of grammar and typos. We would like to apologise for the mistakes that have been now corrected.

Abstract

detec-tion – write detection

Introduction

Line 102, appreciate a better word for proposed

Several fungal species have been proposed to synthetize GT, such as Trichoderma lig-

Methods

Brief about 22 amino acid sequences involved in analysis

As suggested we have now included “After the quick search, we picked 22 homologous GliP sequences belonging to different fungal species”

Line 201

Do all fungal strain sporulate in Sabouraud glucose agar?

Thanks for the observation, we have now included this information “If a species, such as A. lentulus or E. ampelina, could not sporulate in the indicated time or media, a small piece of 1x1 cm of mycelia was sliced from the agar plate and inserted into cell culture flasks”.

Line 206

Czapek-Dox broth was used for all fungal stains?

Yes, and also RPMI 1640

Line 207

Expand RPMI (and HEPES)

Line 273

Italisise A. niger. Some fungal strains throughout the article have not been italicized

All the suggested corrections have been done

Reviewer 4 Report

The manuscript is far from publication because it contains fundamental errors that cannot be rectified through author revisions. The quality of presentation and scientific soundness of the manuscript is quite low.. 

Gliotoxin is a fungal secondary metabolite with impact on health and agriculture. In the manuscript the authors identified homologous gli gene clusters similar to the A. fumigatus GT gene cluster in several fungal species, and only detected GT in Fumigati species and Trichoderma virens during fermentation. In general, the results are reliable and will help investigation in GT producers. However, the manuscript is poorly written with many obvious errors, it is far from publication.

Specific comments:

  1. Some important parameters are missing in “Materials and Methods” part, like the concentration of chloramphenicol in line 201, concentration of Tween-20 in line 204.
  2. HPLC results of culture supernatants and GT/bmGT standards should be included in the manuscript or supplementary file.
  3. In 3.1.2, there are too many contents which belong to “Materials and Methods” part instead of “results” part, they must be removed to “Materials and Methods”.
  4. The authors should carefully check the references, some journal names are presented in full name while the others are in abbreviation.
  5. the figure legend of figures is “GT biosynthesis, secretion and regulation, and role of the genes involved in this process”, but there are no contents about secretion and regulation of GT in figure 1.
  6. As some species like A. lentulus apparently owns all the machinery needed for GT biosynthesis failed to produce GT in the research, it is better for the authors to confirm whether these genes transcript during fermentation in Czapek-Dox and RPMI 1640 broth by RT-PCR. These kinds of clusters are so called cryptic biosynthesis cluster and they are unusually silent in conventional culture conditions.
  7. It’s better for the authors to rearrange the clusters of figure 5 based on the order in figure 3
  8. The problems in scientific names of species are serious in the manuscript. For instance, “C assianum” in line 386 should be “C. assianum”; “spp” in “Aspergillus spp” mustn’t be italic, species names in line 44 and line 45 must be spelled out in full because it is the first time they appear. “Penicillium” in 497 should be italic. There are also problems in writing of gene names, like “gliJ, gliA and gliT” in line 397, they all should be italic. The authors must correct these errors very carefully.

Author Response

We thank to the referee for his comments, constructive criticisms and carefully editing of grammar and typos. We would like to apologise for the mistakes that have been now corrected.

  1. Some important parameters are missing in “Materials and Methods” part, like the concentration of chloramphenicol in line 201,concentration of Tween-20 in line 204.

All the paremeters have been included

  1. HPLC results of culture supernatants and GT/bmGT standards should be included in the manuscript or supplementary file.

As suggested we have now included a supplementary figure (Figure S1) containing GT and bmGT standards chromatogram and GT/bmGT standard calibration curves.

  1. In 3.1.2, there are too many contents which belong to “Materials and Methods” part instead of “results” part, they must be removed to “Materials and Methods”.

As suggested we have now moved several sentences to Materials and methods and other minor modifications were made.

  1. The authors should carefully check the references, some journal names are presented in full name while the others are in abbreviation.

We have carefully checked and corrected all the references.

  1. The figure legend of figures is “GT biosynthesis, secretion and regulation, and role of the genes involved in this process”, but there are no contents about secretion and regulation of GT in figure 1.

Sorry for the mistake, we have now changed the figure legend to “GT biosynthesis and role of the genes involved in this process”

  1. As some species like A. lentulus apparently owns all the machinery needed for GT biosynthesis failed to produce GT in the research, it is better for the authors to confirm whether these genes transcript during fermentation in Czapek-Dox and RPMI 1640 broth by RT-PCR. These kinds of clusters are so called cryptic biosynthesis cluster and they are unusually silent in conventional culture conditions.

Indeed, it would be interesting to see at what level the expression occurs since probably the fact that they do not produce GT could be due to the fact that the gene is not being expressed, in fact our results, having used two different culture media, would confirm this hypothesis.

We have now included in Discussion: “Probably, some species that apparently owns all the machinery needed for GT biosynthesis failed to produce GT in the research due to culture conditions, since it is well known that cryptic genes are repressed in normal culture conditions"

  1. It’s better for the authors to rearrange the clusters of figure 5 based on the order in figure 3

As suggested we have now rearranged the clusters of figure 5 based on the order in figure 3

  1. The problems in scientific names of species are serious in the manuscript. For instance, “C assianum” in line 386 should be “C.assianum”; “spp” in “Aspergillus spp” mustn’t be italic, species names in line 44 and line 45 must be spelled out in full because it is the first time they appear. “Penicillium” in 497 should be italic. There are also problems in writing of gene names, like “gliJ, gliA and gliT” in line 397, they all should be italic. The authors must correct these errors very carefully.

All the suggested corrections have been done

Round 2

Reviewer 4 Report

The authors have made changes as suggested and the revised manuscript seems much better than previous version. The responses from the authors are convincing, so I think this manuscript will be of great interest for Journal of fungi readers.